# OpenReview forum: "Improved Stochastic Optimization of LogSumExp"
_ICLR.cc/2026/Conference — Submitted to ICLR 2026_

### Official Review · Reviewer_f1Yu · 2025-10-27

**Soundness:** 2
**Presentation:** 3
**Contribution:** 2
**Rating:** 2
**Confidence:** 5

**Summary:**

This paper introduces a stochastic optimization framework for objectives involving the LogSumExp function (or log-partition functional), proposing a new “Safe KL” divergence formulation that yields a rescaled SoftPlus approximation of LogSumExp. The authors claim this relaxation preserves convexity, smoothness, and enables efficient stochastic gradient methods applicable to entropy-regularized optimal transport (EOT) and distributionally robust optimization (DRO). The theoretical analysis shows convergence properties and connections to Conditional Value-at-Risk (CVaR). Empirical demonstrations include low-dimensional EOT and DRO on the MNIST and California Housing datasets.

The theoretical formulation is sound and mathematically elegant, but the novelty and experimental validation are limited. The approximation technique and results overlap with prior SoftPlus-based LogSumExp relaxations, and the numerical demonstrations are too small-scale to convincingly support the claimed generality.

**Strengths:**

(1)The theoretical reformulation through an alternative f-divergence is clear and well motivated.
(2)The approach preserves convexity and smoothness, with rigorous proofs of approximation bounds.
(3)Applications to EOT and DRO are reasonable and illustrate numerical stability over the standard LogSumExp formulation with larger stepping.

**Weaknesses:**

(1)The proposed rescaled SoftPlus approximation is not entirely novel. Similar ideas have appeared—for instance, in “Safety Alignment Should Be Made More Than Just a Few Tokens Deep” (arXiv:2406.05946), where the authors also analyze the effect of the scaling parameter \rho.
(2) The paper lacks comparison with other established techniques for optimizing or bounding LogSumExp, such as those discussed in
- Convex Maximization via Adjustable Robust Optimization (https://optimization-online.org/wp-content/uploads/2020/07/7881.pdf)
- LSEMINK: A Modified Newton-Krylov Method for Log-Sum-Exp Minimization” (Kan et al. 2023) deals with optimization of LogSumExp via Newton-Krylov methods (https://arxiv.org/abs/2307.04871)
- Accurate Computation of the Log-Sum-Exp and Softmax Functions (https://arxiv.org/abs/1909.03469)
- The broader review “What is the log-sum-exp function?” (Higham, 2021 https://nhigham.com/2021/01/05/what-is-the-log-sum-exp-function/)
(3) The experimental design is too simple for a top ML conference. For instance, the “semi-continuous” EOT test uses 1D feature inputs, which is insufficient to demonstrate scalability or real-world relevance. Stronger baselines (e.g., multi-dimensional or high-resolution OT problems) should be included.
(4) The empirical section does not explore how the rescaling parameter \rho affects performance and accuracy, especially in the MNIST experiments (Section 3.3). Sensitivity or ablation tests are needed to justify the parameter’s influence on convergence and bias.

**Questions:**

(1) The manuscript claims novelty for the “rescaled SoftPlus approximation” of the LogSumExp (LSE) function. Please clarify precisely how your method differs in formulation, analysis, or practical behavior from existing works, for instance listed above.
(2) Can you include additional experiments and comparison with another optimization methods of LSE, such as piecewise-linear approximations, Newton-type solvers, or surrogate bounds, and provide wall-clock/iteration numbers for a range of dimensions and scaling parameters?
(3) Could you provide a sensitivity/ablation study of \rho in both training stage and final objective quality, along with guidance on how to choose \rho in practice?
(4) How does your method scale to higher feature dimensions (e.g., dim > 10, or dim > 100), larger term-counts in LogSumExp, and different \rho values?

---

> ### Author Response · Authors · 2025-11-20
> **Response to Reviewer Comments**
>
> We thank the reviewer for their feedback.
>
> First, we would like to clarify that our approach, as outlined in the abstract and contributions part, enables the use of **stochastic** gradient methods (SGD) for the log-partition function with a **large or infinite** number of terms. In other words, we propose a general SGD-friendly approach for LogSumExp/log-partition minimization that is efficient across many applications, including, e.g., continuous (infinite dimensional) OT, where stochastic approximation (as opposed to sample average approximation) is essential.
>
> Below are our detailed answers to the reviewer’s comments.
>
> 1. Regarding the provided references:
>     - https://arxiv.org/pdf/2406.05946: this work employs a **different rescaling** of Softplus, i.e., $\frac{1}{\rho} \log(1+ e^{\rho z})$ compared to our $\frac{1}{\rho} \log\left(1 + \rho e^z\right)$, and our approach additionally involves minimization w.r.t. $\alpha$ in (6). Their analysis in Appendix D only considers asymptotic behavior, and neither limit $\rho \to 0$ nor $\rho \to +\infty$ recovers the LogSumExp. In contrast, our approach gives an approximation to LogSumExp/log-partition functional with explicit accuracy for any $\rho \in (0,1)$. Thus, unless we are missing something, that paper is not directly related to ours.
>     - “Convex Maximization via Adjustable Robust Optimization” addresses the different problem of LogSumExp **maximization**, rather than minimization. Unlike our work, it only allows a finite number of components, and it is not clear how this technique can be applied to stochastic optimization (SGD).
>     - LSEMINK is a method that requires computing the **full softmax** function in each iteration, which is incompatible with problems involving a large or infinite number of terms inside the logarithm. It is also limited to objectives involving LogSumExp(Ax+b), i.e., composition of LogSumExp with a **linear** function. Finally, it is fully **deterministic**, whereas our approach supports nonlinear problems, infinite sums, and the stochastic optimization setting required in Sect. 3.
>     - “Accurate Computation of the Log-Sum-Exp and Softmax Functions” analyzes algorithms for **evaluating** (not minimizing) the **full** LogSumExp (and Softmax), which is exactly what we want to avoid in our setup, where the number of exponential terms is extremely large (or infinite).
>     - “What is the log-sum-exp function?” This blog post discusses the same topic as the previous paper.
> In summary, these papers are not directly related to our general approach and cannot be considered as baselines in the applications we consider. The main reason is that they do not consider stochastic optimization (SGD) settings. Nonetheless, we have included them for broader context in the review of related works. The changes in the updated manuscript are marked in blue.
> 2. We agree that additional experiments would be valuable and plan to incorporate them during the discussion period. We note, however, that the original submission already addresses the reviewer's key experimental suggestions:
> Although we use 1D distributions in the EOT experiment, the problem is infinite-dimensional since we treat them as general continuous distributions. Moreover, the sum in the LogSumExp is infinite.
> The feature dimension of the MNIST dataset is 28x28=784, i.e., >100 as suggested.
> The California housing dataset contains 20640 objects (i.e., terms inside the logarithm).
> We also emphasize our theoretical contribution, in particular, the mathematically principled safe KL divergence and its dual.
> 3. We are working on the sensitivity/ablation tests of the rescaling parameter $\rho$ and plan to update the manuscript during the discussion period.
>
> To summarize, we feel that reviewer f1Yu seems to have either misunderstood or overlooked important aspects of our theoretical contributions and experimental evaluation, and the absolutely certain rejection recommendation is based on such misunderstandings. We believe that our clarifications have now made a strong case for our contributions. We humbly ask the reviewer to please consider improving the score.

---

### Official Review · Reviewer_vrxv · 2025-10-31

**Soundness:** 3
**Presentation:** 2
**Contribution:** 4
**Rating:** 8
**Confidence:** 2

**Summary:**

The paper considers the minimization of the functional $f \mapsto \log \int e^f d\mu$, whose specific forms appear in many applications, such as computing softmax probabilities and maximum-likelihood estimation for exponential families. The functional  can be viewed as LogSumExp, whose computation is known to suffer from numerical-stability issues, which the proposed approximation can mitigate substantially as shown emprically in the paper.

**Strengths:**

The paper studies a fundamental question: the optimization of LogSumExp. The contribution is clear in both theory and the provided empirical results, which demonstrate improvements across various tasks. As far as I can follow, I did not find inconsistencies or technical flaws, and thus the contribution appears significant.

**Weaknesses:**

Quantitive summary of the results of the experimental section is missing, especially in Section 3.1. Is it Figure 2? This figure is not referenced anywhere in the text. Due to the unclear presentation, it is difficult to obtain an accurate view of how well the proposed approach performs overall compared to the baseline.

Writing is not always smooth. Some examples:

“First, the decision variable φ(or a parameter θdefining φ) often has large or infinite dimension... The first challenge is usually addressed by the use of first-order methods, especially stochastic gradient descent (SGD), with cheap iterations.” What this means? Is a reader supposed to follow?

Lines 55 and 57: incosistent usage of dashes

**Questions:**

How to pick the parameter $\rho$ controlling the accuracy of the approximation?

---

> ### Author Response · Authors · 2025-11-20
> **Response to Reviewer Comments**
>
> We thank the reviewer for their time, thoughtful comments, and positive evaluation of our work. We address the specific points below.
>
> The reviewer is correct that Figure 2 presents the quantitative results for Section 3.1. The summary of this experiment is provided in lines 324-331 of the updated manuscript. Similarly, the results for Sections 3.2 and 3.3 are summarized in lines 397-404 and 478-485, respectively. These summaries were already present in the original submission, and together they demonstrate the superior performance of our method in the considered setups.
>
> We appreciate the reviewer's feedback regarding lines 46-50 of the introduction. We have revised the explanation to improve its clarity. The inconsistency in the usage of dashes has also been fixed. The changes in the updated manuscript are marked in blue.
>
> The choice of the parameter $\rho \in (0,1)$ involves a trade-off between the smoothness of the approximation and its accuracy. A larger $\rho$ increases smoothness (aiding SGD convergence) but reduces approximation quality, while a smaller $\rho$ does the opposite. Thus, to prioritize training stability with SGD, a moderate value is effective. When approximation fidelity is critical, a smaller $\rho$ is preferable. The selection strategy is application-specific; one option is to perform a hyperparameter search. We are working on additional experiments to illustrate this dependence, which we plan to include in the revision.

---

### Official Review · Reviewer_kNFY · 2025-10-31

**Soundness:** 2
**Presentation:** 3
**Contribution:** 2
**Rating:** 4
**Confidence:** 3

**Summary:**

In this paper,  the authors propose a new strategy of approximating the LogSumExp function which is based on the usage of a modified KL divergence (safe KL) in the dual formulation. This approximation is applied in relevant tasks, including the computation of entropy-regularized optimal transport (EOT) problem. It is shown that the approximation helps to alleviate the overflow issue in these applications.

**Strengths:**

The idea of the paper is rather interesting and supported by the conducted theoretical analysis. The authors show several use-cases of the proposed approximation.

**Weaknesses:**

My major concerns correspond to the overall contribution of the proposed approximation of LogSumExp and its practical significance. While the derived theoretical results show some nice properties of the derived approximation, its experimental validation has some white spots. To begin with, in section 3.1, the authors tested their approximation by introducing it in the method for entropic optimal transport (Genevay et al., 2016) which is rather old while being classic. For example, this method does not allow for the direct computation of EOT plans which is a more important problem than the computation of EOT cost itself. In recent years, many other methods for computing EOT plans between the continuous distributions have been derived, see, e.g., (Seguy et al., 2018; Daniels et al., 2021, Gushchin et al., 2023, Korotin et al., 2023). The paper would benefit from the testing of your approximation in these more recent approaches.

Besides, while in remark 3.1, the authors note that for small value of parameter $\epsilon$  the method of (Genevay et al., 2016) exhibits the overflow issue, the conducted experiment reveals the behaviour of this approach and its modified version (with the proposed regularization) only for one parameter $\epsilon$. Here I kindly suggest the authors perform an ablation study of this parameter (for bigger and smaller values) to show how it affects the performance of the method with and without regularization. Similarly, I kindly suggest the authors perform an ablation study on the parameters $\lambda$ and $\beta$ appearing in denominator of exponent in the distributionally robust optimization problems – balanced (16) and unbalanced (20) –  in experiments from sections 3.2, 3.3.

Moreover, there are several aspects of the paper which could be written more clearly. First, In equation (2), the authors introduce an alternative form of the functional $F(\phi,\mu)$ using the Gibbs variational principle. I kindly suggest including reference to one of the papers where this principle is introduced. Many formulas should be written in a more rigorous manner, i.e, integrals should include the spaces over which they are taken and sup/inf should be taken over some optimization variable (see Eq. (2)). The typos are summarized below.

**Overall**, I have doubts regarding the contribution of the approximation of the LogSumExp function derived in the paper. While the theoretical results support this approximation, its usage should offer considerable benefits over the usage of the initial function. To prove it, some additional experimental evaluations with the recent entropy-regularized OT methods and ablation study on the parameters should be considered.

**Questions:**

- Could you perform an ablation study of the parameters $\epsilon$ in section 3.1 and parameters $\lambda$ and $\beta$ in section 3.2 and 3.3, respectively?
- Could you test your approximation with some of the more recent EOT methods?

*Typos:*

- in Eq. (2): ‘sup’ -> ‘sup_{\nu}’
- in line 108: minimum -> maximum?
- in Eq. (2), (5), (11)-(13) and almost everywhere in the text: \int -> \int_{X}
- how do you calculate the ‘optimality gap’ visualized in Fig. 2?

**References.**

Seguy, V., Damodaran, B. B., Flamary, R., Courty, N., Rolet, A., & Blondel, M. (2018, April). Large-Scale Optimal Transport and Mapping Estimation. In ICLR 2018-International Conference on Learning Representations (pp. 1-15).

Gushchin, N., Kolesov, A., Korotin, A., Vetrov, D. P., & Burnaev, E. (2023). Entropic neural optimal transport via diffusion processes. Advances in Neural Information Processing Systems, 36, 75517-75544.

Daniels, M., Maunu, T. & Hand, P. Score-based generative neural networks for large-scale optimal transport. Advances in neural information processing systems, 34:12955–12965, 2021.

Korotin, A., Gushchin, N., & Burnaev, E. Light Schrödinger Bridge (2024). In The Twelfth International Conference on Learning Representations.

---

> ### Author Response · Authors · 2025-11-20
> **Response to Reviewer Comments**
>
> We are grateful for the reviewer's insightful feedback and constructive suggestions. Below, we address each of the comments in detail.
>
> We thank the reviewer for the suggestion to test our approximation on more recent approaches to EOT, and we are currently working on additional experiments. We would like to clarify that our paper's primary contribution is a **general-purpose approach** for LogSumExp minimization, with EOT as one of the applications (other examples are in Sections 3.2-3.3). We note that methods like (Gushchin et al., 2023; Korotin et al., 2024) are tailored to the quadratic cost, whereas our approach works with a **general cost function**. Furthermore, it leverages a rich duality theory, which also potentially allows for the reconstruction of the transport plan – an important direction for future work. We see significant potential in improving frameworks like (Seguy et al., 2018; Daniels et al., 2021) by using our method with a neural network parameterization instead of an RKHS. Another promising direction for future work is leveraging our approach for minimizing the objective (8) in (Korotin et al.). We thank the reviewer again for these highly relevant references which we included in Appendix B.1 of the revised manuscript.
>
> Thank you for the suggestion regarding ablation studies. We are currently conducting sensitivity analysis on the rescaling parameter $\rho$ and other hyperparameters and plan to include these results in the revised manuscript.
>
> In response to your other helpful comments:
> - We have added two references on the Gibbs variational principle after Eq. (2).
> - All indicated typos have been corrected.
> - The details on the computation of the 'optimality gap' in Fig. 2 were provided in Appendix B.
>
> All changes in the updated manuscript are marked in blue.

---

### Official Review · Reviewer_FHGh · 2025-10-31

**Soundness:** 3
**Presentation:** 3
**Contribution:** 2
**Rating:** 4
**Confidence:** 4

**Summary:**

The paper introduces a novel and mathematically sound approximation for the LogSumExp function, termed "safe KL divergence," which is designed for efficient and stable stochastic optimization. The method preserves desirable properties like convexity, and its approximation error is explicitly controlled by a tunable parameter. The empirical results in continuous optimal transport and distributionally robust optimization are strong, demonstrating clear practical benefits in terms of convergence speed and numerical stability over the **chosen** baselines. However, the paper's central claim regarding its novelty is significantly overstated due to a major omission of relevant literature on existing unbiased estimators for the log-partition function, which directly contradicts the assertion that "no cheap unbiased stochastic gradient have been proposed."

**Strengths:**

1. A key strength is that the bias introduced by the approximation is not arbitrary. The paper provides clear theoretical bounds showing the error is of order O(ρ), allowing practitioners to explicitly trade off accuracy for computational stability (Proposition 2.4 and Corollary 2.5).
2. The approximation preserves essential properties like convexity and smoothness, which are crucial for guaranteeing the performance of gradient-based optimizers.
3. The resulting gradient estimator is simple to implement and does not require complex machinery like control variates or sophisticated sampling schemes, making it an attractive and easy-to-adopt tool.

**Weaknesses:**

1. The paper does not do justice to prior work on approximating the log partition function. As of now, the prior work is largely incomplete. Few examples:
https://arxiv.org/pdf/2004.00353
https://arxiv.org/pdf/1703.05160
https://arxiv.org/pdf/1306.4032
https://projecteuclid.org/journals/statistical-science/volume-30/issue-4/On-Russian-Roulette-Estimates-for-Bayesian-Inference-with-Doubly-Intractable/10.1214/15-STS523.full
https://arxiv.org/pdf/1703.07370
https://arxiv.org/abs/1901.10517
2. Please add a proof for Lemma 2.7 into the Appendix.
3. Empirical baselines are weak (see item 1 above).

**Questions:**

1. Could the authors compare with existing unbiased estimators like REBAR or those based on Russian Roulette truncation?
2. Any other baselines Authors would like to add?
3. Why fρ(t) is strongly convex (lines 218, 219)?

---

> ### Author Response · Authors · 2025-11-20
> **Response to Reviewer Comments**
>
> We are grateful to the reviewer for their time, effort, and valuable suggestions. We have carefully studied the provided references, and our comments on each are below.
> 1. Russian Roulette truncation (RRT) in https://arxiv.org/pdf/2004.00353 is specialized to latent variable models, which are out of scope for our paper. Moreover, unlike our approach, it is not clear whether RRT preserves convexity and smoothness, or has explicitly controllable nonasymptotic approximation error. Additionally, as mentioned in the paper, the proposed RRT estimator either has infinite variance or infinite expected number of sampled points. This makes it difficult to obtain reasonable theoretical guarantees, unlike our proposed approximation.
> 1. https://arxiv.org/pdf/1703.05160 focuses on estimating the partition function in the particular case of log-linear models. Unlike our paper, it is unclear whether this approach can be applied to optimization involving the partition function and beyond the log-linear setting.
> 1. https://arxiv.org/pdf/1306.4032 focuses on Bayesian Inference which is not the focus of our work. The main question in that paper is estimating the partition function itself, and unlike our paper, it is not clear whether this approach can be applied to optimization involving the partition function and whether it preserves convexity and smoothness, or has explicitly controllable nonasymptotic approximation error.
> 1. REBAR in https://arxiv.org/pdf/1703.07370 is also designed for discrete latent variable models that we do not consider in our paper. Moreover, unlike our approach, it is not clear whether REBAR preserves convexity and/or smoothness or has explicitly controllable nonasymptotic approximation error.
> 1. https://arxiv.org/pdf/1901.10517 focuses on subset sampling, and it is not clear whether this approach can be applied to optimization involving the partition function that we consider in our paper.
>
> To sum up, these references are tailored to specific tasks, and it is not clear whether these techniques can be applied to general optimization problems, whether they preserve convexity and smoothness, or have explicitly controllable nonasymptotic approximation error like our proposed approach. Nonetheless, for a broader context, we included these papers in the review of related works in the introduction. Given the above, it is also not clear how to empirically compare our approach with the ones in the references provided by the reviewer, as those approaches seem to be designed for different tasks than the ones we consider. We are grateful to the reviewer for bringing up papers on latent variable models. Our approximation can be potentially used for such models, and this is an interesting direction of future work.
>
> In the case that we misunderstood your comment, we would be extremely grateful if the reviewer could provide further comments on how the approaches in these papers can be used for general minimization problems that involve functional (1) that we consider in our paper.
>
> We have added a proof for Lemma 2.7 to the Appendix. The changes in the updated manuscript are marked in blue.

---

### Author Response · Authors · 2025-12-03
**Summary of the paper updates**

We would like to thank once again the Reviewers and Area Chairs for their work.

To address the reviewers' comments, in summary, we made the following updates to the paper.

1. We fixed all the typos and added additional clarifications where the reviewers asked.
2. We extended the related literature review by the references provided by the reviewers for a broader context.
3. We reworked Sect. 3.1 with applications to entropic optimal transport (eOT). Specifically, we considered MNIST and EMNIST-letters datasets, and compared the proposed approach to the LSOT framework Seguy et al. (2018) as suggested by one of the reviewers. We also discussed other recent OT solvers, positioning our approach relative to them. The new experiments also include sensitivity tests with respect to the parameters $\varepsilon$ and $\rho$ as suggested by reviewers.
4. We moved the previous experiments on eOT comparing our method to Genevay et al. (2016) to Appendix B.
5. We revised Sect 3.2 with applications to distributionally robust optimization with KL divergence by including the results on the sensitivity of our approach to the approximation parameter  $\rho$ and temperature $\lambda$, as suggested by the reviewers.
6. While the revised paper already demonstrates robustness of the proposed approximation to $\rho$ and other parameters, we are finalizing additional experiments for Sect. 3.3 with applications to distributionally robust optimization with unbalanced OT. If the paper is accepted, we will be able to include them in the camera-ready version.

All changes in the updated manuscript are marked in blue. We believe that with these changes, we have resolved all the reviewers’ concerns.

---

### Meta-Review · Area_Chair_2La7 · 2025-12-28

**Summary:**

This paper considers the optimization of the LogSumExp objective function. The idea is to replace the KL-divergence by a safe KL-entropy term in the Gibbs variational principle of the LogSumExp function. The paper derives the approximation and smoothness properties of the rescaled SoftPlus formulation, while preserving convexity. The approximation to LogSumExp can be solved via stochastic gradient methods. Applications to entropic optimal transport (EOT) and distributionally robust optimization (DRO) are demonstrated.

The reviewers raised several common weakness and questions:

- novelty of the rescaled SoftPlus approximation;
- lack of comparison with prior techniques for optimizing or bounding LogSumExp;
- empirical baselines are weak.

**Reviewer Concerns:**

In response, the authors clarified the differences between their work and the prior papers pointed out by the reviewers. The authors also added numeric results to accommodate some of the reviewers’ requests, which were helpful to show the sensitivity of the proposed method in terms of the rescaling parameter, entropic regularization parameter, and temperature parameter.

**Reviewer Scores:**

4 reviewers submitted their comments and scores (8/4/4/4) with confidence (2/3/5/4), with average score 4.5 and average confidence 3.5.

The positive score was less constructive and confidence level is relatively low, compared with other reviewers. All other negative scores raised substantial concerns, and I was not sure if those concerns were confidently addressed to justify improved scores.

---

### Decision · Program_Chairs · 2026-01-26

Reject